# Landscape Ecological Risk Assessment and Planning Enlightenment of Songhua River Basin Based on Multi-Source Heterogeneous Data Fusion

Ying Zhao [1], Zhe Tao [1,2], Mengnan Wang [3], Yuanhua Chen [4], Rui Wu [1] and Liang Guo [1,2,*]



1   School of Environment, Harbin Institute of Technology, Harbin 150090, China
2   State Key Laboratory of Urban Water Resource and Environment, Harbin Institute of Technology,
    Harbin 150090, China
3   College of Information Science and Engineering, Shandong Agricultural University, Taian 271000, China
4   Eco-Product Development Research Center Co., Ltd., China Energy Conservation and Environmental
    Protection Group, Beijing 100084, China
*   Correspondence: guoliang0617@hit.edu.cn

**Abstract:** In this study, the remote sensing images of the 30 km buffer zone from Zhaoyuan to Baidu of the Songhua River, which is rich in land use types and frequent in human activities, were selected as the research object to analyze land use change and driving factors. The objective of this research is to evaluate the ecological risk of watershed landscapes and provide a basis for watershed ecological environment protection and planning. On this basis, the landscape pattern index data were extracted, and a three-dimensional comprehensive index system of the natural, social and landscape pattern was constructed. In addition, based on the spatial principal component analysis (SPCA), data fusion was carried out to improve the accuracy and comprehensiveness of landscape ecological risk assessment results. The risk level of watershed landscape ecology was divided into low ecological risk, medium-low ecological risk, medium ecological risk, medium-high ecological risk, and high ecological risk by the Natural Breaks method. According to the results of the landscape ecological risk assessment and the characteristics of risk sources in each risk level area, the ecological protection and planning enlightenment suitable for each risk level area were obtained. The research content can provide ideas and evidence for environmental managers to formulate ecological risk protection strategies and reduce the impact of ecological risk threats.

**Keywords:** Songhua River Basin; multi-source data fusion; remote sensing image processing; land use; landscape ecological risk

## 1. Introduction

In recent years, the disturbance to the Songhua River Basin ecosystem has gradually increased due to excessive reclamation and human activities. The ecological space of forest land has shrunk, the landscape has been fragmented, the biodiversity has been threatened, and the stability of the river basin ecosystem has decreased [1,2]. Landscape ecology is a new branch of environmental science, geography, and ecology [3–5]. It can be used to study the relationship between ecological processes and regional ecosystems under different landscape scales, spatial patterns, and policy organizations [6,7], and can also reflect the impact of different landscape characteristics on the spatial pattern changes and ecological effects of ecosystems [8]. From the landscape scale perspective, landscape ecological risk assessment comprehensively considers the influences of natural and human factors on regional ecological risk. At the same time, it also examines the dynamic changes in different spatial characteristics and ecological risks of regional ecosystems caused by changes in landscape heterogeneity under various driving factors [9,10]. Landscape ecological risk assessment can reflect the stress factors of ecosystem services and judge the degree of disturbance of ecosystem functions and structures [11,12].





Some progress has been made in landscape ecological risk assessment at home and abroad. Jin et al. evaluated the landscape ecological risk of the Delingha urban area on the Qinghai Tibet Plateau and obtained the change law of ecological risk in the plateau [13]. Asef et al. jointly built a landscape ecological risk assessment model through a neural network, Markov chain, and ecological connectivity index, assessed the ecological environment of the urban area of Bojnourd, a northern city in Eastern Iran, and found that the landscape ecological function in this area was seriously degraded [14]. Hossein et al. analyzed the ecological risks of different landscape ecology and land use on the content and distribution of heavy metals in the coastal sediments of the Persian Gulf [15]. Based on the ecological risk assessment of the river basin landscape, the interference degree of human activities on the river basin landscape can be determined at a high level. Then, the rational planning and healthy development of the river basin landscape ecology can be promoted [16–18]. Zhang et al. took the Harbin section of Songhua River Basin as the research area, constructed the landscape ecological risk index by combining the landscape vulnerability index and the proportion of landscape type area with the AHP weight assignment method, and finally studied the spatial and temporal change characteristics and spatial autocorrelation of landscape ecological risk in this area [19]. Taking the Manas River Basin as the research area, Kang et al. selected the Landsat remote sensing images of 2000, 2005, 2010, and 2015. Based on the landscape pattern index, they used geostatistics to explore the landscape ecological risk degree and spatial-temporal differentiation characteristics of the Manas River Basin [20].

Landscape ecological risk is the negative impact of the interaction between human social activities, natural factors, and the landscape pattern on the ecological environment [21–23]. In previous studies, the selections of landscape ecological risk assessment index systems were relatively simple, and most of them only considered the impact of the landscape pattern index. Multi-source data fusion is the comprehensive application of various types of data information on the same screen, absorbing the characteristics of different data sources and extracting more unified, better, and richer information than single data [24–26]. If multi-source data fusion can be realized in ecological environment management, more comprehensive and reasonable ecological evaluation results will be obtained [27]. At present, some scholars have applied data fusion technology to the field of the ecological environment [28–30]. Multi-source heterogeneous data mainly refer to structured and unstructured data [31,32]. However, the application of data fusion in landscape ecological risk assessment is very few, and no similar application has been found in this research field.

In the study of the water environment and ecological risk in the basin, the traditional research method obtains structured monitoring data by sampling and analyzing the water environment and ecological environment based on sites and locations. Due to the single degree of data structure and discontinuous temporal and spatial distribution, it is difficult to effectively display the comprehensive information of the basin environment and ecology. With the popularization of remote sensing data and technology, the application of remote sensing technology to watershed monitoring has unique advantages. Compared with traditional sites, remote sensing data can obtain comprehensive spatial information of the watershed, which makes it is easier to achieve the whole watershed coverage. The data acquisition time is short, and the detection sustainability is stronger. This method comprehensively considers other types of data, realizes the combined application of unstructured data and structured data, and makes up for the single data structure problem in the environmental field.

Based on high-resolution remote sensing images, this study extracts land use types and landscape pattern data of the Songhua River Basin and combines multi-source data such as the water environment and the nature and social economies to achieve effective analysis of unstructured data to semi-structured data and structured data, which complements the current application of single structured data in the environmental field. This study analyzed the possible relationship of water environment change in the basin from the perspective of

the macro non-point source landscape pattern. Compared with the traditional ecological assessment methods, this study evaluates the heterogeneity of ecological risks in the watershed buffer zone through the three dimensions of the nature society landscape pattern, providing a basis for watershed ecological protection and planning. Based on the landscape pattern of high-resolution remote sensing images and taking into account the impact of natural factors and socio-economic factors on the risk level, it can more comprehensively reflect the interference of human activities on the watershed water environment and ecosystem, which is an important basis for promoting the watershed water environment protection and the healthy development of landscape ecology.

## 2. Materials and Methods

### 2.1. Study Area

In this study, the remote sensing images of the 30 km buffer zone from Zhaoyuan to Baidu of Songhua River, which is rich in land use types and frequent in human activities, were selected as the research object. The boundaries of the original buffer zone were adjusted to facilitate the extraction and analysis of remote sensing images. The upper and lower boundaries of the study area were buffer zone boundaries. The left boundary was at the Sancha estuary of Zhaoyuan, and the buffer zone was cut off in part of Jilin Province. The right boundary was the boundary between Tonghe County and Fangzheng County. The overview of the research scope in this study is shown in Figure 1.

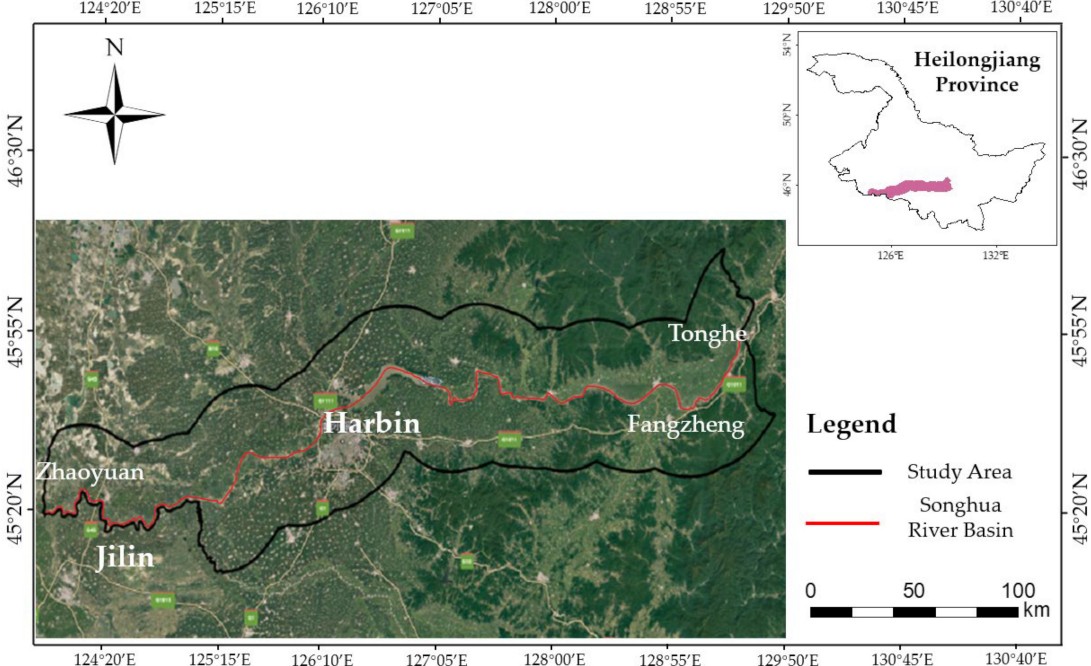

**Figure 1.** The overview of the research scope.

Cultivated land was the main land type in the buffer zone of the studied river basin, with an average area of 11,790 km². The built-up land was mainly located in the urban area of Harbin and other counties, cities, and towns. Most of the forest land was distributed in the east of the buffer zone of the study basin, including the Mongolian mountain range, the remaining Lesser Khingan Mountains range, and the northwestern section of the Changbai Mountain branch of Zhang Guangcai Ling. The average annual runoff of the Zhaoyuan–Harbin section of the Songhua River was $3.627 \times 10^{10}$ m³, the average annual rainfall was 343 mm, the average temperature in summer was 22.3 °C, and the elevation of the buffer zone in the river basin was mostly between 70 m and 185 m. The river basin buffer zone included Zhaoyuan County of Daqing City, Zhaodong County of Suihua City, the main urban area of Harbin City, Bayan County, Bin County, Yilan County, Tonghe County, Mulan

County, and Fangzheng County. In 2018, the total population of these areas was 9.2 million. In 2018, the regional GDP of Harbin, Heilongjiang Province, was 630.05 billion yuan.

*2.2. Dataset and Landscape Ecological Risk Assessment Method*

2.2.1. Data Sources

Landsat 5 and Landsat 8 remote sensing data were selected as the data sources in this study. The data were from the official website of the United States Geological Survey. The selected remote sensing images were clear and partly cloudy, which could meet the needs of manual visual interpretation. Landsat 5 TM remote sensing image data were used for remote sensing data from 2005 to 2011, and Landsat 8 OLI remote sensing image data were used for remote sensing data from 2013 to 2018. The time range of remote sensing images was from May to September, totaling 42 remote sensing images. The Landsat 8 satellite was launched in 2013, constituting time series data with Landsat 5.

The landscape ecological risk assessment considered three dimensions, and the data sources of the three dimensions were as follows.

(1)     Natural dimension indicator data.

In the natural dimension, three natural indicators were selected: elevation, soil type, and soil texture. The soil type raster data, soil texture raster data, and elevation raster data were obtained from the Resource and Environmental Science and Data Center, Chinese Academy of Sciences. The above data types included structured data such as tables and unstructured data such as maps.

(2)     Social dimension indicator data.

The social and economic data from 2005 to 2018 (population and GDP data, etc.) were all taken from the statistical yearbook of Heilongjiang Province and the statistical yearbook of Harbin. The raster data of spatial population distribution, the raster data of GDP per unit area, and the vector data of administrative divisions were obtained from the Resource and Environmental Science and Data Center, Chinese Academy of Sciences.

(3)     Landscape pattern dimension indicator data.

The normalized difference vegetation index (NDVI) of the landscape pattern dimension was from the Resource and Environmental Science and Data Center, Chinese Academy of Sciences.

2.2.2. Extraction of Land Use Types

The visual interpretation method was used to classify the features based on the differences in the spectra, geometry, spatial distribution, and textures of different features in remote sensing images. According to the principles of land use type classification in China and considering the needs of this study, land use types in the river basin buffer zone were identified as seven categories: water, forest land, dry land, bare land, built-up land, water field, and saline land [33,34]. In this study, classification standards of land use types are shown in Table 1.

**Table 1.** Land use type interpretation signs in the buffer zone.

| Number | Land Use Type | Interpretation Sign | Image Display |
|---|---|---|---|
| 1 | Water | Its geometric boundary is clear and distinct, smooth. The color is dark blue. |  |
| 2 | Forest Land | It is mostly found in mountainous areas and has a clear trend. The color is dark green. |  |

**Table 1.** *Cont.*

| Number | Land Use Type | Interpretation Sign | Image Display |
|--------|---------------|---------------------|---------------|
| 3 | Dry Land | It has a regular and continuous distribution with different spectral characteristics and is smooth. The color is orange. |  |
| 4 | Bare Land | It is distributed at the top or bottom of the mountain, and its texture is rough. The color is brownish. |  |
| 5 | Built-up Land | It is planar in distribution, with a rough texture and distinct borders. The color is red. |  |
| 6 | Water Field | It is distributed near the river, lumpy. The color is light green. |  |
| 7 | Saline Land | It shows a planar distribution, distributed on the edge of the bare ground of the pond with obvious borders. The color is grayish-white. |  |

Based on the interpretation signs in Table 1, the training samples for land use type classification were created using remote sensing image processing software. To judge whether any two features in the remote sensing image could be classified effectively, the training sample resolution was calculated. The need was considered satisfied if the classification sample resolution took a value between 0 and 2.0. This study chose the maximum likelihood classification method to classify remote sensing images, a supervised classification method. It was assumed that the number of pixels of each land use type in each band conformed to the normal distribution rule. The likelihood of the image elements belonging to different land use types in the training set was calculated, and the specific category of the image elements was determined according to the maximum likelihood principle. After performing the same operation on all image elements, the land use type classification was completed.

2.2.3. Accuracy Verification

In this study, the confusion matrix evaluation method was used to verify the accuracy of land use type results from remote sensing images. The assessment was performed by comparing the classification results of some image elements in the remote sensing classification map with the real classification of that image element in reality. The overall accuracy (OA) is the number of correctly classified pixels divided by the total number of pixels involved in the comparison and expressed as a percentage. The Kappa coefficient is one of the indicators of image classification accuracy, and its value is between 0 and 1.

In this study, 130 sites with unchanged land use types were selected from the remote sensing images from 2005 to 2018. The actual land use types were determined through field investigation, and precision evaluation was conducted. The evaluation results are shown in Table 2. The overall average classification accuracy is 81.58%, and the Kappa coefficient is about 0.7–0.8, indicating that the classification results of this remote sensing image are good and meet the needs of subsequent research and analysis.

**Table 2.** Overall classification accuracy and Kappa coefficient of remote sensing images.

| Time | Overall Classification Accuracy (%) | Kappa Coefficient |
|------|-------------------------------------|-------------------|
| 2005 | 85.73 | 0.8113 |
| 2006 | 80.65 | 0.7642 |
| 2007 | 79.81 | 0.7218 |
| 2008 | 82.94 | 0.8073 |
| 2009 | 76.18 | 0.7186 |
| 2010 | 82.25 | 0.7752 |
| 2011 | 82.47 | 0.7762 |
| 2012 | 84.33 | 0.8013 |
| 2013 | 85.40 | 0.8065 |
| 2014 | 83.80 | 0.7916 |
| 2015 | 80.33 | 0.7592 |
| 2016 | 75.28 | 0.7122 |
| 2017 | 83.21 | 0.7863 |
| 2018 | 79.84 | 0.7546 |

*2.3. Landscape Ecological Risk Assessment Method*

2.3.1. Index Selection of Landscape Ecological Risk Assessment

The study constructed a three-dimension comprehensive index system of nine influencing factors of the natural, social and landscape pattern. It was used to evaluate the heterogeneity of ecological risks and the impact of integrated natural–human effects in the river basin.

(1)    Natural dimension indicators.

In the natural dimension, three natural indicators were selected: elevation, soil type, and soil texture. Elevation, as a terrain element, could impact geological disasters such as land erosion and landslides. The carbon sequestration capacity and erosion resistance of soils had important implications for regional climate regulation, soil and water conservation, and food production. Different soil types and textures in soil carbon sequestration capacity and erosion resistance were different. The stronger the carbon sequestration capacity and the higher the erosion resistance, the lower their ecological risk level.

(2)    Social dimension indicators.

Population density and GDP per unit area were chosen as evaluation indicators in the social dimension. The higher the population density, the more frequent the human activities and behaviors, and the higher the ecological risk intensity. Similarly, the larger the value of GDP per unit area, the higher the regional economy level, the greater the modification and impact on the ecological environment, and the higher the intensity of ecological risk.

(3)    Landscape pattern dimension indicators.

Land use type, NDVI, Shannon's evenness index (SHDI), and Contagion Index (CONTAG) were selected in the landscape pattern dimension. Land use types were rated: built-up land was level 5, bare land was level 4, dry land and water field were level 3, saline land was level 2, and water and forest land were level 1. The higher the level, the higher the ecological risk. The NDVI was used to reflect the status of the land cover vegetation, and its value ranged from $-1$ to $1$. The values of NDVI were close to 0 for buildings, negative for water bodies, and positive for vegetation. The raster data of the SHEI and CONTAG were calculated using the moving window method. The SHEI can indicate the evenness of the distribution of different landscape types and the maximum landscape diversity at a given landscape richness. The more evenly distributed the landscape, the higher the richness and the higher the stability of the ecosystem. The CONTAG describes the degree of clustering or the extension trend of different plate land classes in the landscape, which contains spatial information and is one of the most important indicators to describe the landscape pattern. A higher value indicates that a certain patch type in the landscape has

formed good connectivity and the ecosystem is more stable. The lower value indicates that the landscape is a dense pattern with multiple elements, a higher degree of fragmentation of the landscape, and a weaker anti-interference ability. In this study, for subsequent ecological risk modeling, various indicators were normalized, and the units of each indicator were unified.

2.3.2. Landscape Ecological Risk Assessment Method Based on SPCA

The SPCA method can correspond to a matrix for each variable. At the same time, the principal component factor analysis results can be clearly implemented in each raster corresponding to the space, and the original principal component analysis results can be visually extended to the two-dimensional space [35–37]. It has a good spatial visualization effect. In this study, the SPCA method would be used to evaluate the landscape ecological risk, and the formula was as follows [38]:

$$R = \sum_{i=1}^{m} \sum_{j=1}^{n} (a_{ij} F_j) \tag{1}$$

where $R$ is a comprehensive landscape ecological risk assessment result, $a_{ij}$ is the $j$th principal component corresponding to the $i$th raster, and $F_j$ is the eigenvalue contribution rate of the $j$th principal component.

In this study, based on the calculation principle of the SPCA, the original nine influence factors (i.e., land use type, SHEI, CONTAG, NDVI, population density, GDP per unit area, soil type, soil texture, and elevation raster data) were input into the remote sensing image processing software in turn. The generated nine principal components were calculated, and the eigenvalues, eigenvectors, and contribution rates of each principal component were obtained.

It is generally considered that the cumulative contribution rate of more than 90% is statistically significant [39]. Therefore, several principal components satisfying the condition that the cumulative contribution rate exceeds 90% were selected as the final results from the above nine principal components. This study assumed that $k$ ($k < 9$) principal components met the requirements. With the remote sensing image processing software as a tool, the scores of $k$ principal components were calculated from the original variable data. The equation (Equation (1)) of river basin landscape ecological risk was used as the evaluation index, combined with the contribution rate of the eigenvalues of each principal component for weighted superposition to calculate the comprehensive evaluation results of integrated landscape ecological risk. The results were divided into five levels by the Natural Breaks method: low ecological risk, medium-low ecological risk, medium ecological risk, medium-high ecological risk, and high ecological risk.

## 3. Results and Discussion

### 3.1. Extraction of Land Use Types

This study used remote sensing image processing software to complete the pre-processing of remote sensing images. Based on the differences in the spectrum, texture, geometry, and spatial distribution characteristics of different features in remote sensing images, the land use types along the Songhua River Basin were divided into seven categories by visual interpretation: water, forest land, dry land, bare land, built-up land, water field, and saline land.

In this study, based on the interpretation signs of land use types along the Songhua River Basin, the training set of land use type classification was constructed, and the maximum likelihood supervised classification method was used to classify land use types. The ROI separable tool was used to evaluate the different degrees of each type of land use types in the training set. After multiple assessments and adjustments, the resolution values of all samples were above 1.8, and most of them were greater than 1.9, which indicated that the samples could be effectively classified and meet this study's needs. In this study, the confusion matrix evaluation method was used to verify the accuracy of land classification,

and 130 points with unchanged land types in remote sensing images from 2005 to 2018 were selected. The actual land use types were determined through on-the-spot investigation, and the accuracy was evaluated by comparing them with the results extracted from remote sensing images. The overall accuracy of this research could reach about 80%. At the same time, the accuracy corresponding to the kappa coefficient was at the level of "good" (0.7–0.8) or even "very good" (0.8–1.0). Therefore, the extracted land use type could be studied and analyzed accordingly. The land use type map of the 30 km buffer zone of the Zhaoyuan–Harbin section of Songhua River in 2018 is shown in Figure 2.

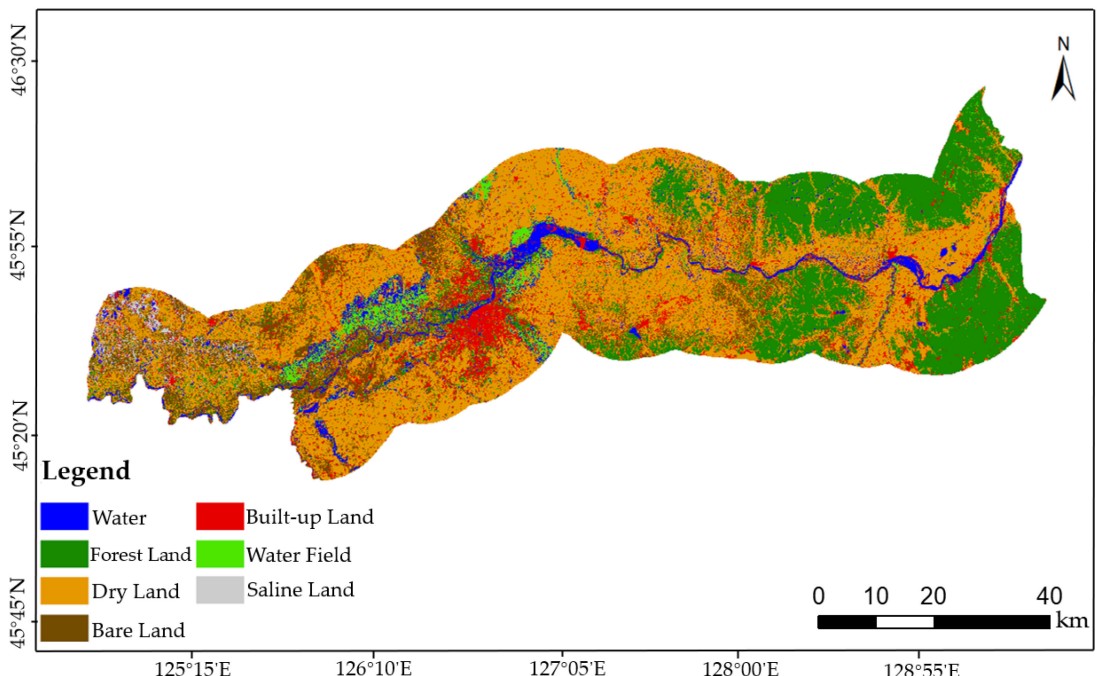

**Figure 2.** Land use type map of 30 km buffer zone in Harbin section of Zhaoyuan–Harbin Songhua River in 2018.

### 3.2. Analysis of Land Use Evolution in River Basin

3.2.1. Analysis of Land Use Area Change

The extracted data on land use type were statistically analyzed in this study. Figure 3 shows the bar chart of land use type area in the 30 km buffer zone of the Zhaoyuan–Harbin section of the Songhua River Basin from 2005 to 2018.

From 2005 to 2018, the main land use types in the buffer zone were dry land, forest land, bare land, and built-up land, with the total proportion of the three being 92.96%. Dry land was the most important land type in the 30 km buffer zone of the Zhaoyuan–Harbin section of Songhua River Basin, accounting for the largest proportion in the study area, reaching 47.87%. Forest land was the second largest land type in the study area, with an average of 24.40%, which played an important role in soil and water conservation in the river basin. The area of forest land in the buffer zone showed a rising upward trend, with an annual average increase of about 78.1 km$^2$. The function of water and soil conservation in the buffer zone has been gradually improved, which has positively impacted the improvement of the ecological water environment. The area of bare land showed a downward trend, with an annual average decrease of about 127.57 km$^2$. With the social and economic development, from 2005 to 2018, the regional GDP of Harbin increased from CNY 179.64 billion to CNY 630.05 billion, and the regional GDP of built-up land grew from CNY 16.77 billion to CNY 60.94 billion. Against this background, the area of built-up land in the buffer zone showed a stable growth trend, with an average annual growth of about 107.47 km$^2$.

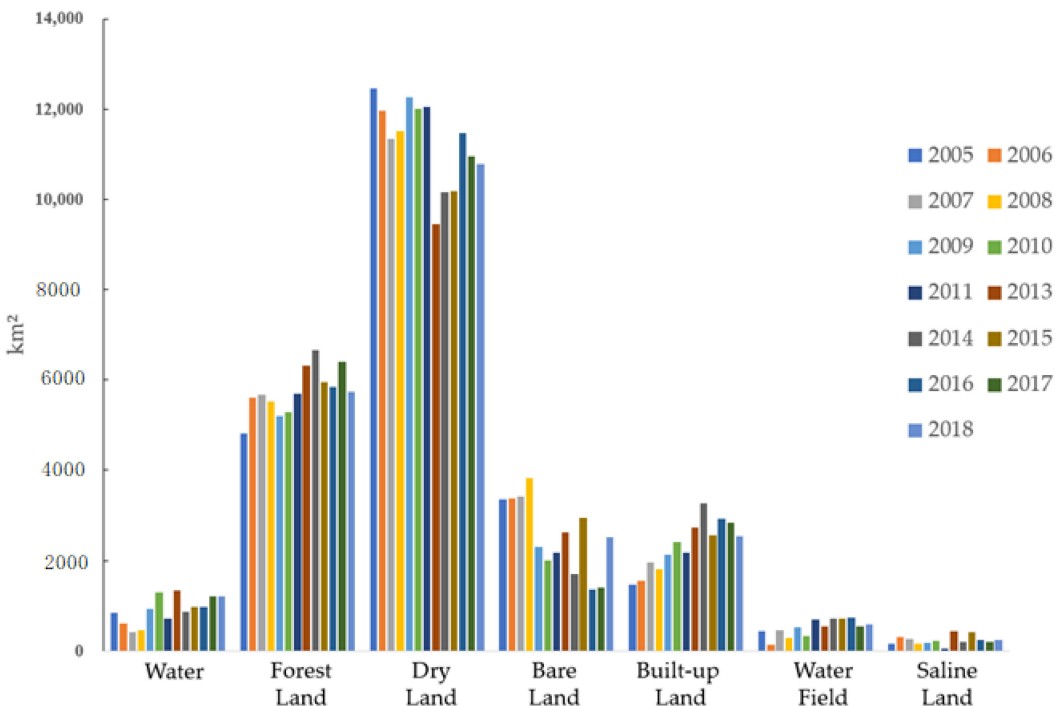

**Figure 3.** Land use type area map of the river basin buffer zone, 2005–2018.

3.2.2. Analysis on Spatial Transfer of Land Use Types

In this study, to analyze the spatial transfer of land use types in the buffer zone of the river basin more intuitively and clearly, the land use type areas with equal time intervals were evenly selected, and we chose land use type data in 2005, 2010, 2014, and 2018. The spatial transfer matrix of land use types was calculated using remote sensing image processing software, and the spatial transfer maps of land use types in 2005–2010, 2010–2014, and 2014–2018 were obtained (Figures 4–6).

In 2005–2010, 2010–2014, and 2014–2018, the dry land in the buffer zone of the river basin was transformed into built-up land, bare land, and forest land, but the distribution of land use types that transferred from dry land to forest land was not consistent. From 2005 to 2010, the conversion of dry land to forest land was mainly distributed on both sides of the Songhua River bank and near the foothills within the watershed buffer zone in the Zhaoyuan County area of Daqing City. From 2010 to 2014, the types of land use transferred from dry land to forest land were more concentrated in the Acheng District of Harbin City and the foothills of the northwest section of Zhang Guangcai ridge, the branch of Changbai Mountain in Bin County. In the analysis of spatial transfer intensity of land use types in the river basin buffer zone, the temporal analytic hierarchy process showed that from 2010 to 2014, the land use types in the river basin buffer zone switched with each other most frequently. The intensity of land use change was faster, and the impact of human activities was greater. The geographic analytic hierarchy process showed that the transfer in and out of land use types of forest land and dry land within the river basin buffer was relatively stable at all times and was not susceptible to external disturbances.

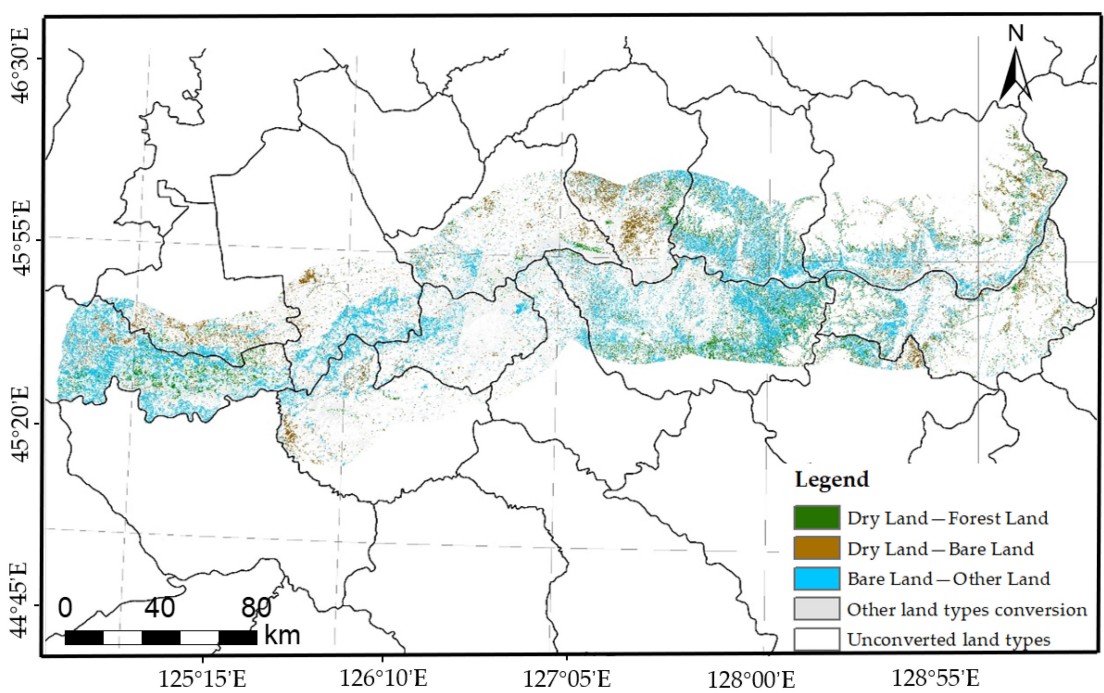

**Figure 4.** Spatial transfer map of land use types in the buffer zone of the river basin, 2005–2010.

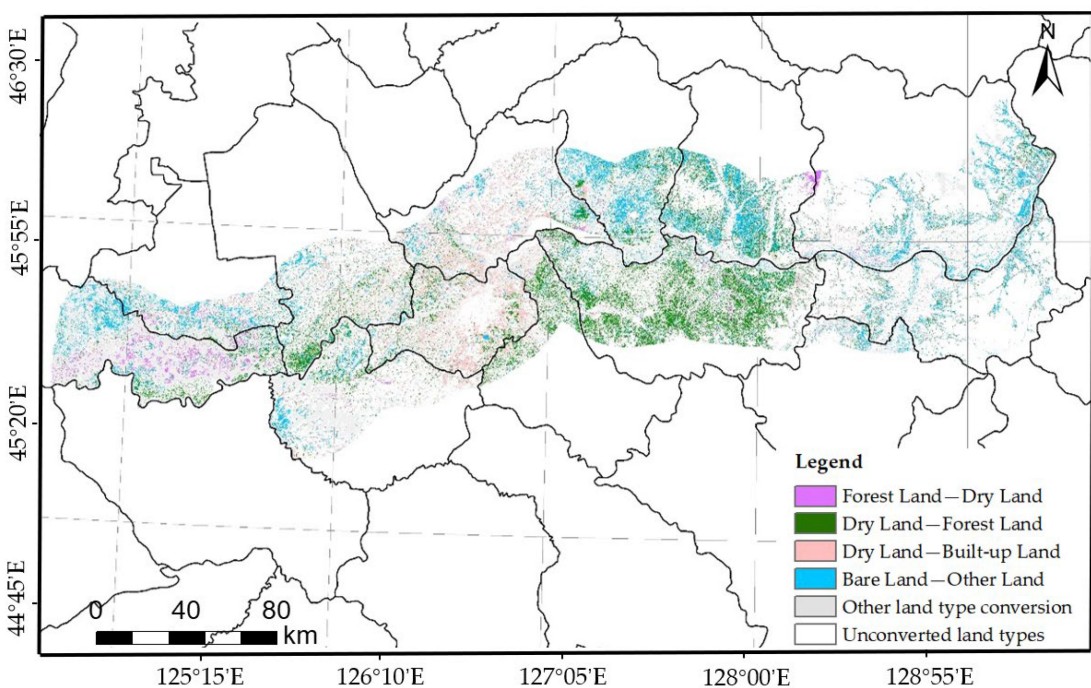

**Figure 5.** Spatial transfer map of land use types in the buffer zone of the river basin, 2010–2014.

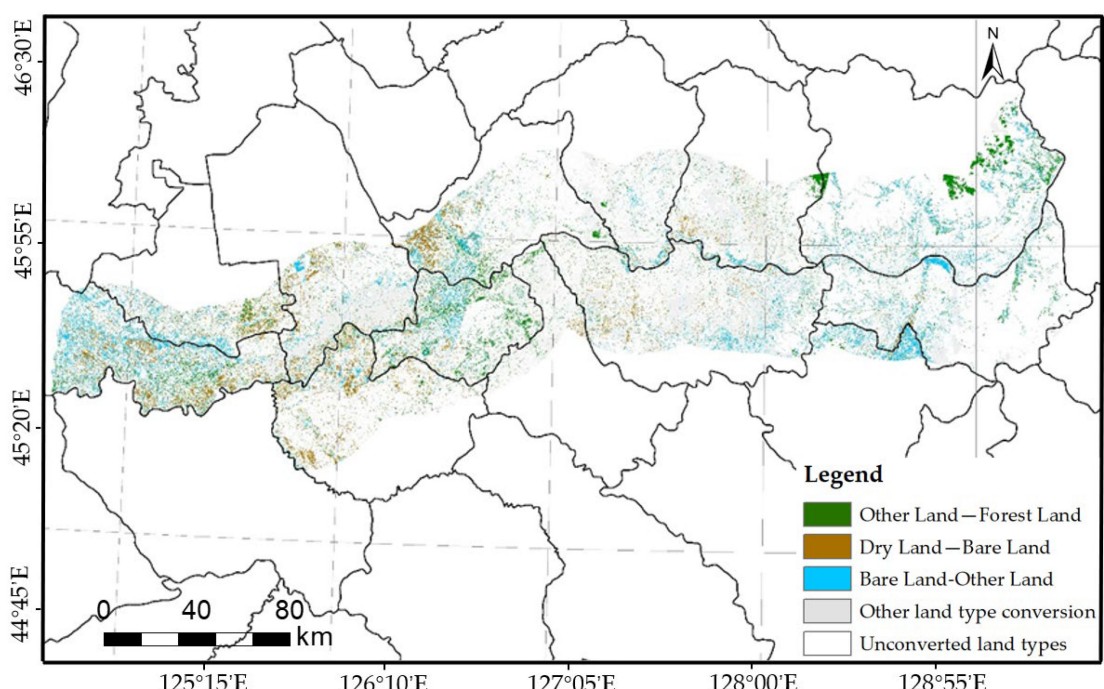

**Figure 6.** Spatial transfer map of land use types in the buffer zone of the river basin, 2014–2018.

3.2.3. Analysis on Driving Factors of Land Use Change

This section analyzes the driving factors of land use change in the 30 km buffer zone from Zhaoyuan to Harbin in the Songhua River Basin from 2005 to 2018 from natural and socio-economic driving factors.

(1)　Natural driving factors

Climate, hydrology, and terrain mainly analyzed the natural driving factors. There was a close relationship between climate change and the growth of vegetation. In this study, the selected remote sensing image data were all summer data, and the time was taken from May to September. From 2005 to 2018, the annual average summer precipitation of Harbin, Daqing, and Suihua was 318.6 mm, 339.4 mm, and 370.9 mm, respectively. During this period, the summer precipitation showed an overall upward trend. From 2005 to 2018, the annual average summer precipitation of Harbin, Daqing, and Suihua increased by 6.7 mm, 6.0 mm, and 10.4 mm, respectively. The interannual increase in summer precipitation had a potential deep-seated impact on the increase in water area, forest land area, and water field area in the buffer zone of the river basin. The temperature affected the water and vegetation in the buffer zone of the Songhua River Basin. The change in temperature affected the surface water evaporation, which further affected the water level, the water area, and the vegetation growth. From 2005 to 2018, the average summer temperatures in Harbin, Daqing, and Suihua were generally stable, and the annual average summer temperatures were 22.6, 22.6, and 21.9 °C, respectively. From 2005 to 2018, the annual runoff of the Harbin section of the Songhua River's main stream showed an overall upward trend, and the annual runoff increased by about $1.39 \times 10^9$ m$^3$. The increase in the runoff of the Songhua River's main stream in the river basin's buffer zone had a direct impact on the increase in the water area. At the same time, it also affected the change in land use types near the Songhua River bank in the buffer zone, including the land for built-up land of the dam, water fields, dry land, and forest land. The terrain was an important reason for the distribution of land types in the buffer zone of the river basin, which affected the concentration process of surface runoff and the distribution of groundwater. In this study, the terrain was relatively stable, with little variation in the study time scale.

(2)　Socio-economic driving factors

This study area contained many towns, such as the main city of Harbin, etc. Socio-economic factors had a more significant impact on the buffer zone of the Songhua River Basin, especially at smaller time scales. This section was analyzed mainly in terms of demographic and economic factors.

Population growth and reduction played an important role in driving land use change. In this study, the resident population of the main urban areas of Harbin, Bin County, Yilan County, Bayan County, Mulan County, Tonghe County, Fangzheng County, Zhaoyuan County and Zhaozhou County of Daqing City, and Zhaodong County of Suihua City which were covered by the buffer zone of the basin were counted. The population was the highest in 2009 (9.803 million) and gradually decreased from 2010 to 2018, with a population of 9.2042 million in 2018. The agricultural population of Harbin District and County covered by the river basin buffer zone reached the highest in 2011, with a population of 3.8261 million, then gradually decreased to 3.68 million in 2018. The urban population of Harbin District and County covered by the river basin buffer zone increased from 4.132 million in 2005 to 4.2554 million in 2016 and then decreased to 4.1301 million in 2018. The decrease in the total population and the agricultural population and the increase in the urban population led to the reduction in the amount of farmland converted into built-up land.

The regional GDP was the comprehensive performance of regional economic development. The areas with high GDP had more investment in urban infrastructure, which directly affected the change in land use types. Harbin's GDP increased from CNY 179.64 billion in 2005 to CNY 630.05 billion in 2018. The primary and secondary industries reached the maximum in 2016, reaching CNY 69.12 billion and CNY 189.67 billion, respectively. The built-up land in the secondary industry increased from CNY 16.77 billion in 2005 to CNY 60.94 billion in 2018. The proportion of the tertiary industry increased from 50.1% in 2005 to 64.8% in 2018. In the study area, the overall rise in economic level and the optimization of the industrial structure positively impacted the conversion of land use types. The built-up land in the buffer zone of the river basin increased steadily, from 1473.63 km$^2$ in 2005 to 2539.33 km$^2$ in 2018.

From the perspective of territorial spatial planning, among the three strategic patterns of the main functional area planning in Heilongjiang Province, a "one center and two wings" urban strategic planning pattern had been formed, with Harbin as the center and Qiqihar and Daqing as the support cities. It promoted the development of urban spatial patterns and had certain positive factors in improving the construction land area in the river basin. The agricultural products supplied by Songnen Plain and Sanjiang Plain formed a strategic agricultural pattern of "three zones and five belts", which promoted the construction of an industrial belt based on corn, soybeans, livestock products, and potatoes in the agricultural product zone of the Songnen Plain. It maintained the agricultural space based on the permanent basic farmland red line and played a certain role in protecting and maintaining the dry land in the buffer zone of the river basin. The forest land ecological functional area of the Greater and Lesser Khingan Mountains, the forest land ecological functional area of Changbai Mountain, and the wetland ecological functional area of Sanjiang Plain created the "two mountains and one plain" ecological security strategy pattern. It was related to the growth of the forest land area in the southern part of the remaining veins of the Lesser Khingan Mountains and the northwest section of the Changbai Mountain branch of Zhang Guangcai Ling in the buffer zone of the study area.

*3.3. Risk Assessment Results and Analysis*

3.3.1. Results of the SPCA

In this study, based on the calculation principle of the SPCA, remote sensing image processing software was applied to calculate the principal components of the original nine data variables, and a total of five principal component factors were obtained under the condition that the cumulative contribution rate exceeded 90%. The eigenvalues and cumulative contribution rates of each principal component are shown in Table 3. The

loadings of the original evaluation factors corresponding to each principal component are shown in Table 4.

**Table 3.** The eigenvalues and cumulative contribution rates of each principal component.

| Principal Component | Eigenvalues | Contribution Rates | Cumulative Contribution Rates |
|---|---|---|---|
| pc1 | 3.451 | 41.7% | 41.7% |
| pc2 | 2.846 | 26.7% | 68.4% |
| pc3 | 2.152 | 10.1% | 78.5% |
| pc4 | 1.617 | 8.1% | 86.6% |
| pc5 | 1.359 | 5.8% | 92.4% |

**Table 4.** Load matrix of each principal component.

| Comprehensive Evaluation Factors | Original Index | pc1 | pc2 | pc3 | pc4 | pc5 |
|---|---|---|---|---|---|---|
| Landscape pattern factors | Land use types | 0.792 | 0.567 | 0.111 | 0.185 | 0.024 |
| | SHDI | 0.496 | 0.526 | 0.099 | 0.555 | 0.305 |
| | CONTAG | 0.297 | 0.627 | 0.263 | 0.588 | 0.191 |
| | NDVI | 0.126 | 0.075 | 0.718 | 0.100 | 0.604 |
| Social factors | Population density | 0.031 | 0.020 | 0.136 | 0.368 | 0.003 |
| | GDP per unit area | 0.107 | 0.089 | 0.112 | 0.097 | 0.045 |
| Natural factors | Soil types | 0.125 | 0.042 | 0.068 | 0.416 | 0.316 |
| | Soil texture | 0.009 | 0.006 | 0.600 | 0.313 | 0.635 |
| | Elevation | 0.081 | 0.021 | 0.098 | 0.144 | 0.046 |

According to the analysis of load value results in Table 4, among the three dimensions, the load values of the four indicators of the landscape pattern dimension in the principal component were relatively large, while the load values of the social dimension and the natural dimension were small. It indicated that the landscape pattern factors had a greater influence on the ecological risk assessment results within the 30 km buffer zone of the Zhaoyuan to Harbin section of the Songhua River Basin. Among the natural factors, soil type had a larger loading value (0.416) in principal component 4, indicating that soil types among the natural factors had a greater influence on the assessment results. In the social factors, the load value of population density in principal component 4 was larger (0.313). It indicated that the population density greatly influenced the assessment results.

3.3.2. Ecological Risk Assessment of The River Basin Landscape

The comprehensive landscape ecological risk assessment results were obtained by weighted superposition of five principal component scores. The results showed that the overall risk index value of the buffer zone was between 0.166 and 0.838. The results were divided into five levels by the Natural Breaks method: low ecological risk (with risk indicators 0.166–0.314), medium-low ecological risk (with risk indicators 0.314–0.406), medium ecological risk (with risk indicators 0.406–0.472), medium-high ecological risk (with risk indicators 0.472–0.564), and high ecological risk (with risk indicators 0.564–0.838). The map of the spatial distribution of landscape ecological risks in the 30 km buffer zone of the Zhaoyuan–Harbin section of the Songhua River Basin is shown in Figure 7.

(1)  Regional assessment of low ecological risk level

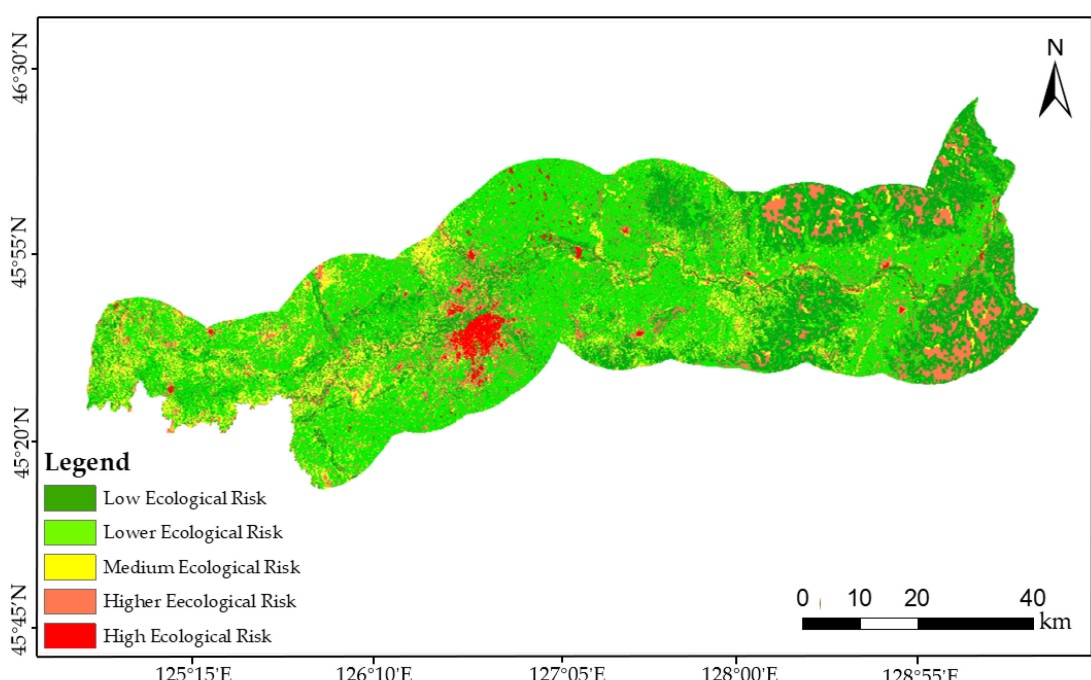

**Figure 7.** The map of spatial distribution of landscape ecological risks in the buffer zone.

In this study, the range value of low ecological risk indicators was between 0.166 and 0.314, with an area of about 6171.8 km$^2$, accounting for 26.3%, which was the second highest in the overall risk of the study buffer zone. As could be seen in Figure 7, the low ecological risk areas were mainly distributed in the mountain areas in the eastern part of the study buffer zone, including the Mongolian mountain range areas in Bayan and Mulan counties of Harbin City, the southern part of the remaining Lesser Khingan Mountains range in Mulan and Tonghe counties of Harbin City, and the northwestern section of the Changbai Mountain branch of Zhang Guangcai Ling in Bin and Founder counties of Harbin City. The NDVI of this area was very high, which was about 0.9. The elevation mainly ranged from 189 m to 450 m, and the main land use type was forest land. In the study area, it played a positive role in soil and water conservation. At the same time, this area had higher ecological stability and a stronger ability to resist risk.

(2)    Regional assessment of medium-low ecological risk level

In this study, the range value of medium-low ecological risk indicators was between 0.314 and 0.406, with an area of about 11,106.0 km$^2$, accounting for the highest proportion of 47.32%. The medium-low ecological risk areas were widely distributed in the plain area, including Zhaodong County of Suihua City, Zhaoyuan County of Daqing city, and all districts and counties except the main urban area of Harbin. The main land use type in these areas was dry land, and the NDVI was higher, about 0.8. The value of the SHEI was lower, and its range was between 0.3 and 0.4. The landscape distribution was relatively single, and the richness was low. The average value of CONTAG in this area was 69.3. It indicated that the plaque types in the medium-low risk areas were good, with higher connectivity and stability. In conclusion, the overall anti-ecological risk capacity of the area was higher.

(3)    Regional assessment of medium ecological risk level

In this study, the range value of medium ecological risk indicators was between 0.406 and 0.472, with an area of about 2768.0 km$^2$, accounting for the fourth ranking, reaching 11.79%. The medium ecological risk areas were mainly distributed on both sides of the Songhua River. The land use type was bare land, and the soil type was the sand bar of the river. The NDVI was low, about 0.3. The medium ecological risk areas were also distributed

in Zhaoyuan County, Daqing City. The land use types were mainly saline and bare land, and the NDVI was low, about 0.3. The SHEI was between 0.3 and 0.5 at the middle level. The landscape distribution uniformity was general. The average value of CONTAG was 44.3. The overall landscape fragmentation was higher, and the anti-interference ability was weaker. The medium ecological risk areas were also distributed in the west of Zhaodong County of Suihua City, the north of Shuangcheng District of Harbin City, and the east of Hulan District. The area was transformed from dry land to bare land from 2014 to 2018. The NDVI was very low, about 0.2, and the erosion resistance was weaker. The last part was distributed in the Mongolian foothills of Bayan County and Mulan County in Harbin and the foothills of the northwestern section of the Changbai Mountain branch of Zhang Guangcai Ling in Bin and Founder counties of Harbin City. In these areas, bare land accounted for 83.6% of land use types, with an average elevation of 179 m and a certain slope.

(4)　Regional assessment of medium-high ecological risk level

In this study, the range value of medium-high ecological risk indicators was between 0.472 and 0.564, with an area of about 3045.5 km$^2$, accounting for the third ranking, reaching 12.98%. The medium-high ecological risk areas were mainly distributed in the urban area of Harbin and the surrounding small- and medium-sized villages and towns. The land use type was mainly built-up land. It was loosely distributed compared with the dense built-up land in the high ecological risk area. The NDVI was 0, and the soil type was grade 5 urban area with the highest risk level. The average value of SHEI was 0.21, and the landscape diversity was very low with a single distribution. The CONTAG was between 28.3 and 41.7, the overall connectivity of the landscape was poorer, and the fragmentation was higher. The other parts of the medium-high risk areas were mainly concentrated in the high-altitude mountain range areas, which were located in the southern part of the remaining Lesser Khingan Mountains range in Mulan and Tonghe counties of Harbin City and the northwestern section of the Changbai Mountain branch of Zhang Guangcai Ling in Bin and Founder counties of Harbin City. The average elevation of this area was 667 m. The SHEI was between 0 and 0.16, which was very low. This might be due to the single landscape type in this area, only forest land, and lower overall landscape diversity. Due to the higher altitude of the area, the soil had a weaker erosion resistance and carbon sequestration capacity, the landscape type was single, and the diversity was lower. Therefore, the ecosystem was more vulnerable to disturbance and had a higher ecological risk rating.

(5)　Regional assessment of high ecological risk level

In this study, the range value of high ecological risk indicators was between 0.564 and 0.838, with an area of about 379.2 km$^2$, accounting for the least, which was 1.62%. The high ecological risk areas were concentrated in the urban areas of Harbin and other counties and towns with dense areas of built-up land. The NDVI, CONTAG, and SHEI were all close to 0, and the population density was about 11,000 people per km$^2$. This area had the weakest anti-interference ability and the highest ecological risk level.

*3.4. Ecological Protection and Planning Enlightenment*

Based on the results of landscape ecological risk assessment and the characteristics of risk sources in each risk level area, different ecological risk protection measures should be taken to reduce the impact of ecological risk threat factors. Through comprehensive analysis, the specific ecological protection and planning enlightenment of each risk level area in the study area were obtained, which provided important auxiliary information for scientific management of the ecological environment.

The high ecological risk areas had typical distribution characteristics. They were concentrated in the built-up land-dense areas, including the Harbin urban area and other urban built-up land-dense areas. The areas with high population density and human activities were most affected by human behavior. Therefore, the appropriate population

capacity of the city needs to be planned reasonably so that the number of people in the city is lower than the appropriate population capacity to ensure healthy and sustainable development. Effective planning and management can be carried out in the urban core area, including using innovative environmental protection technologies and improving resource conservation and environment-friendly social construction. We should appropriately increase investment in environmental protection in urban areas, improve environmental protection infrastructure in each city, and actively promote new energy vehicles to reduce urban energy use and build clean cities. The CONTAG and SHEI of high ecological risk areas were close to 0, the landscape types were single, and the overall diversity of the landscape was almost 0. Therefore, it is necessary to make rational planning for urban land use, appropriately plant trees, increase shrub and grassland areas, build more urban ecological parks, increase urban landscape types and diversity, improve the anti-interference ability of ecological risks in urban areas, and establish an ecological harmonious urban environment.

Some of the medium-high ecological risk areas were concentrated around the core urban area of Harbin and small- and medium-sized villages and towns. Some industrial enterprises were distributed around the core urban area of Harbin. The relevant departments should focus on treating enterprises and factories with larger pollution, effectively manage and supervise the discharge of enterprises, and effectively rectify the enterprises whose discharge does not meet the standards to ensure that the industrial discharge in this area meets the standards. Currently, the construction of sewage treatment facilities around the city and small- and medium-sized villages and towns is not perfect, and the domestic wastewater in some areas is discharged dispersedly. The relevant departments should appropriately strengthen domestic wastewater treatment facilities in these areas and improve the degree of centralized sewage treatment. The average value of SHEI in this area was 0.21, the landscape diversity was very low, and the distribution was single; the CONTAG was between 28.3 and 41.7, the overall connectivity of the landscape was poorer, and the fragmentation was higher. We should increase the proportion of forest land in suburban areas, plan and build suburban forest land and wetland parks, and strengthen ecological restoration. Another area of medium-high ecological risk was distributed in the high-altitude mountain area of the river basin buffer zone, which was densely forested and far from the human activity areas. However, due to the high altitude, weak soil erosion resistance and carbon sequestration capacity, single landscape type, and low diversity, the ecosystem was vulnerable to natural disasters, including landslides, flash floods from heavy rainfall, and fires caused by dead branches and leaves in dry climates. It is necessary to strengthen the monitoring of the areas and enhance the early warning and emergency response capabilities of natural disasters.

Some medium ecological risk areas were distributed on both sides of the Songhua River. The land use type was bare land, and the soil type was the sand bar of the river. Under the basic premise of ensuring flood control, navigation, and irrigation of the Songhua River, according to the characteristics of hydrology, terrain, and environment of the river basin, the construction of the river buffer zone shall be strengthened, appropriate widths of buffer zone shall be selected, and shrubs and grasslands shall be adopted reasonably to enhance the ecological function, flood control, and landscape function of the river buffer zone and improve the stability of the ecosystem of the river buffer zone. The medium ecological risk areas also included many land use types, where dry land was changed into bare land. The NDVI was about 0.2, the vegetation was poor, cultivation trace gradually disappeared, and soil erosion resistance and anti-ecological risk capacities were weakened. To prevent more dry lands from rising from the medium-low ecological risk level to the medium ecological risk level, it is necessary to strengthen the testing of agricultural land, including farmland yield, soil fertility, pollutant content, etc. For farmland with serious ecosystem disturbance, microbial restoration can be performed, including microbial improvements to enhance soil biological performance and microbial fertilizers. It is also possible to increase soil organic matter and improve soil physical and chemical properties

and biological activities by planting high-quality green manure with a strong nitrogen fixation ability to promote the restoration of the farmland ecosystem and avoid further increase in its ecological risk level.

The low ecological risk area reached 47.32%, which was widely distributed in plain areas. In this area, the NDVI was about 0.8, and the SHEI was between 0.3 and 0.4; the average value of CONTAG was 69.3. The landscape distribution was relatively single, and the richness was low. The plaque types in the medium-low risk level areas were good, with higher connectivity and stability. The buffer zone in the river basin was mainly dry land, with developed agriculture, high fertilizer application, and serious surface source pollution problems. Experts should be hired to scientifically guide farmers in farming, ensure reasonable use of fertilizers and pesticides, reduce farming pollution, and promote the development of green agriculture. The relevant departments should increase the construction of composting facilities to compost livestock and poultry manure to reduce the pollution of livestock and poultry farms while realizing the recycling of resources. Given the relatively dry climate in Northeast China, they should promote agricultural water-saving technology. Micro-irrigation technology can be selected to promote water absorption in the roots of crops and reduce the loss of irrigation resources.

## 4. Conclusions

(1) The application of study on driving factors of land use change complements the deficiency of the application of unstructured image data in the ecological environment. At the same time, it can better promote the application of big data technology in the field of the ecological environment and make up for the shortcomings of existing research.

(2) The evaluation system of multi-source data fusion can improve the accuracy and comprehensiveness of landscape ecological risk assessment results and provide an effective basis for watershed ecological protection and planning. At the same time, the integration of multi-source data can better promote the application of big data technology in the field of ecological environment and make up for the deficiencies of existing research. In the landscape ecological risk assessment, multi-source data were fused, and the landscape pattern index data extracted from remote sensing images were used. The influences of natural and social factors on the landscape ecological risk were comprehensively considered, a three-dimensional comprehensive index system of nine influencing factors of the natural, social and landscape pattern was constructed, and the SPCA method was used to evaluate the landscape ecological risk in the study area comprehensively.

(3) The landscape ecological risk assessment method based on spatial principal component analysis (SPCA) is effective and has good replicability of applications in other regions. The comprehensive landscape ecological risk assessment results were obtained by a weighted superposition of five principal component scores. The results showed that the overall risk index value of the buffer zone was between 0.166 and 0.838. The results were divided into five levels by the Natural Breaks method: low ecological risk, medium-low ecological risk, medium ecological risk, medium-high ecological risk, and high ecological risk. In this study, the spatial distribution characteristics of each ecological risk area and the characteristics of each index were analyzed and evaluated, providing basic information for mining the influencing factors of environmental ecological risk.

(4) In future research, based on the results of landscape ecological risk assessment and the characteristics of risk sources in each risk level area, the ecological protection and planning enlightenment suitable for each risk level area can be obtained effectively. Thus, future research can provide ideas and evidence for environmental managers to formulate ecological risk protection countermeasures and reduce the impact of ecological risk threat factors.

**Author Contributions:** Conceptualization, Y.Z. and L.G.; methodology, Y.Z.; software, M.W.; validation, Y.C. and R.W.; formal analysis, Z.T.; investigation, R.W.; resources, Y.C.; data curation, Z.T.; writing—original draft preparation, M.W.; writing—review and editing, Z.T.; visualization, M.W.; supervision, L.G.; project administration, Y.Z.; funding acquisition, Y.Z. All authors have read and agreed to the published version of the manuscript.

**Funding:** This research was funded by National Natural Science Foundation of China, grant number 71671050.

**Data Availability Statement:** The datasets generated during and/or analyzed during the current study are available from the corresponding author on reasonable request.

**Conflicts of Interest:** The authors declare no conflict of interest.

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
