# Peer review of "Landscape Ecological Risk Assessment and Planning Enlightenment of Songhua River Basin Based on Multi-Source Heterogeneous Data Fusion"

_water, doi:10.3390/w14244060_

Round 1
Reviewer 1 Report
This study takes the typical area of Songhua River as an example, based on the results of the landscape ecological risk assessment and the characteristics of risk sources in each risk level area, the ecological protection and planning enlightenment suitable for each risk level area were obtained, which can provide ideas and evidence for environmental managers to formulate ecological risk protection countermeasures. This is a work of practical significance. The manuscript needs to be improved to be published in Water.
Abstract: The abstract should be comprehensive. It should contain brief introduction, objectives, methodology, niche (novelty, research problem) and summary of significant findings. The abstract needs to be reorganized.
“3. Analysis of Land Use Change in River Basin” and “4. Landscape Ecological Risk Assessment and Planning Enlightenment”, It is suggested that as two sections, put it in "3. Results and discussion ".
3.2.3. Analysis on Driving Factors of Land Use Change : This section analyze the driving factors of land use change in the 30 km buffer in the Songhua River Basin from natural and socio-economic driving factors. What is the relationship between the natural and socio-economic? for driving factors of land use change, whether there is a relative proportion of contribution.
4.1.2. Ecological Risk Assessment of The River Basin Landscape: The results were divided into five levels by the Natural Breaks method. The overall risk index value for each level needs to be given.
Conclusion:This is still a general description. The recommended conclusions focus on the characteristics and accuracy of Multi-source Heterogeneous Data Fusion, as well as replicability of applications in other regions. What needs to be improved in the future research? It is suggested that the author sort it out one by one.
Overall, I think the paper can be accepted after minor revision.

Reviewer 2 Report
It is a practical option to employ remote sensing image processing technology in ecological risk assessment, especially at large scale. This study attempted to implement ecological risk assessment on a part of Songhua River watershed using multi-source data and data fusion. Overall, the MS was relatively well organized. The scientific theme was also clearly stated. However, there are some points in the MS which should be reconsidered by authors as follows. Besides, the language could be further polished.
1. The objective of this study ought to be specified in the abstract and in the introduction.
2. It is better to describe the detailed criteria for discriminating results of ecological risk assessment.
3. ‘the real classification of that image element in reality’ ought to be stated clearly.
4. There might be some confusions if the ‘lower risk’ is higher than the ‘low risk’. The same issue is also for the ‘high’ and ‘higher’ risks.
5. The accuracy verification between the proposed method and the reality might be lack in the MS.
Round 2
Reviewer 2 Report
The issues proposed before were addressed by the authors. The current version the MS has been improved. After the language is further polished up. this MS could be considered for publication.
